# Proposed Mechanisms of Cell Therapy for Alzheimer’s Disease

**DOI:** 10.3390/ijms252212378

**Published:** 2024-11-18

**Authors:** Ekaterina Belousova, Diana Salikhova, Yaroslav Maksimov, Vladimir Nebogatikov, Anastasiya Sudina, Dmitry Goldshtein, Aleksey Ustyugov

**Affiliations:** 1Research Centre for Medical Genetics, Moscow 115522, Russia; ekaterina.belousova.2017@gmail.com (E.B.); diana_salikhova@bk.ru (D.S.); yaroslavmax@mail.ru (Y.M.); sudyina-ak@rudn.ru (A.S.); dvgoldshtein@gmail.com (D.G.); 2Research Institute of Molecular and Cellular Medicine of the Medical Institute Peoples’ Friendship, University of Russia, Moscow 117198, Russia; 3Institute of Physiologically Active Compounds at Federal Research Center of Problems of Chemical Physics and Medicinal Chemistry of the Russian Academy of Sciences, Chernogolovka 142432, Russia; vnebogatikov@gmail.com

**Keywords:** Alzheimer’s disease, β-amyloid, stem cells, neuroprotection

## Abstract

Alzheimer’s disease is a progressive neurodegenerative disorder characterized by mitochondria dysfunction, accumulation of beta-amyloid plaques, and hyperphosphorylated tau tangles in the brain leading to memory loss and cognitive deficits. There is currently no cure for this condition, but the potential of stem cells for the therapy of neurodegenerative pathologies is actively being researched. This review discusses preclinical and clinical studies that have used mouse models and human patients to investigate the use of novel types of stem cell treatment approaches. The findings provide valuable insights into the applications of stem cell-based therapies and include the use of neural, glial, mesenchymal, embryonic, and induced pluripotent stem cells. We cover current studies on stem cell replacement therapy where cells can functionally integrate into neural networks, replace damaged neurons, and strengthen impaired synaptic circuits in the brain. We address the paracrine action of stem cells acting via secreted factors to induce neuroregeneration and modify inflammatory responses. We focus on the neuroprotective functions of exosomes as well as their neurogenic and synaptogenic effects. We look into the shuttling of mitochondria through tunneling nanotubes that enables the transfer of healthy mitochondria by restoring the normal functioning of damaged cells, improving their metabolism, and reducing the level of apoptosis.

## 1. Introduction

Learning and memory are complex cognitive processes that are affected by aging and neurodegenerative disorders, such as Alzheimer’s disease (AD). AD is a progressive brain disorder that causes memory impairment and difficulty in performing daily tasks. It is the most common form of age-related dementia and is characterized by the loss of cholinergic neurons, deposition of proteins, and formation of plaques and tangles [1]. Many therapies for AD have not been successful in improving cognitive function, due to a lack of understanding of the underlying mechanisms and the need for long-term treatment [2,3]. The complexity of AD treatment lies in the need for more research and development of new approaches [4].

One of the main disadvantages of drug therapy for neurological conditions is the lack of stimulation for the regeneration of damaged neurons. This lack of stimulation can lead to reduced cell viability, which in turn inhibits drug transport [5]. Despite the importance of neurotrophic factors for maintaining synaptic plasticity and memory, the signaling mechanisms behind these factors are not fully understood [6]. Insulin therapy may be a safe and short-term intervention to delay cognitive decline, but it does not always lead to significant clinical results [7]. Low-intensity laser therapy holds promise for preventing cognitive impairment by altering brain cell function and metabolic pathways. However, the parameters for personalized treatment have not been determined, and the duration of effects and probability of recurrence remain unclear [8]. A focus on mitochondrial respiration may be effective in the treatment of Alzheimer’s disease due to changes in mitochondrial calcium flow in the early stages of the disease. However, the causes of these changes are not well understood [9]. Additionally, moderate physical activity and an appropriate diet have been associated with a reduced risk of neurodegeneration. However, these methods are not as effective as other treatments for AD [10].

A growing body of research supports the potential of regenerative medicine as a treatment for neurodegenerative diseases. Stem cell therapy has been actively studied for more than 30 years as one of the applications for AD treatment. It offers several advantages over other approaches, as it can increase the level of functional recovery in the central nervous system [11]. By administering stem cells exogenously, depleted nerve pathways can be regenerated, leading to the replacement of damaged nerve cells through the differentiation of transplanted cells [12]. Additionally, administered cells produce various factors, that, in turn, help slow down the process of tissue deterioration and facilitate the recovery of the microenvironment in the injured central nervous system as well as support regeneration [13]. Microglia play an important role in the pathogenesis of Alzheimer’s disease [14]. They are activated in response to the accumulation of beta-amyloid and tau proteins, which can lead to inflammatory processes. When activated, microglial cells secrete inflammatory cytokines, which increase neuroinflammation and may contribute to the progression of the disease [15]. Cell therapy can have an immune-modulating effect by reducing excessive activation of microglia and, therefore, reducing the levels of inflammatory cytokines [13]. It has been reported that the levels of at least four pro-inflammatory mediators, interleukin-1β (IL-1β), interleukin-6 (IL-6), tumor necrosis factor alpha (TNFα), and prostaglandin E2 (PGE2), have decreased [16]. This therapy also has the potential to eliminate neurofibrillary tangles and promote mitochondrial transport, improving cognitive functions [17,18]. The purpose of this review is to explore various types of stem cells, including neural stem cells, glial progenitor cells, mesenchymal stem cells, and embryonic stem cells, as well as induced pluripotent stem cells, in the context of their potential application for Alzheimer’s disease treatment. Particular emphasis is placed on understanding the mechanisms by which these stem cells may be used in AD therapy.

## 2. Stem Cells in AD Therapy

The following types of stem cells are currently being used to treat Alzheimer’s disease: neural stem cells (NSCs), mesenchymal stem cells (MSCs), embryonic stem cells (ESCs), and induced pluripotent stem cells (iPSCs) [19]. The most effective treatments for AD focus on targeting the disease’s pathology at an early stage in order to preserve cerebrovascular function.

### 2.1. Neural Stem Cells

As neural stem cells (NSCs) play a significant role in brain homeostasis and repair, they have shown potential for the treatment of Alzheimer’s disease in the early stages. These cells can replace damaged neurons and astrocytes, and they also exert various paracrine effects that help to maintain brain function [20]. In 2018, McGinley et al. found that human NSC transplantation improved cognitive abilities in APP/PS1—a mouse model of AD with a mutation in the amyloid precursor protein and presenilin 1 [21]. The transplantation was performed in the fimbria–fornix area and significantly improved cognitive function over 4 and 16 weeks, as assessed by memory tests. In addition, Hayashi et al. transplanted both human- and mouse-derived NSCs into mice with AD, and both types of cells showed positive results [22]. More recent studies have focused on understanding the cellular mechanisms of NSC action and their therapeutic potential in AD. Apodaca et al. found that extracellular vesicles from human NSCs can reduce the signs of the disease [23]. They injected 2- and 6-month-old mice with extracellular vesicles from 5xFAD mice. This treatment significantly reduced the accumulation of β-amyloid (Aβ) plaques in both age groups, demonstrating a neuroprotective effect on AD neuropathology. In 2022, Revuelta et al. studied microglia-mediated inflammation and NSC differentiation in AD, as well as the possible therapeutic effect of blocking the K(V)1.3 channels [24]. They found that K(V)1.3 blockers inhibited microglia-induced neurotoxicity by reducing the expression of pro-inflammatory cytokines through the NF-κB and p38 MAPK pathways. The administration of NSCs to mice improved their performance in learning and memory tasks, such as the Morris water maze and context-dependent object recognition [25]. One of the main drawbacks of the NSC therapy is its low viability. In many cases, the loss can be as high as 90%. As a result, NSC replacement requires additional measures to ensure stem cell survival [26,27]. Overall, NSC therapy showed great efficacy in treating AD at an early stage [28,29,30].

### 2.2. Glial Progenitor Cells

Another potential strategy for the treatment of AD is stem cell transplantation, which involves the differentiation of stem cells into glial cells, specifically astrocytes. Astrocytes play a crucial role in the regulation of the central nervous system (CNS) and are highly secretory cells that can release hundreds of molecules involved in nerve tissue function. These molecules are essential for normal CNS function, but abnormal regulation of their secretion has been linked to certain CNS disorders. Recent studies have highlighted the importance of astrocyte signals in regulating their functions in both normal brain function and in disease. A more detailed analysis of the secretome of astrocytes could lead to a better understanding of their full potential, not only in pathological events but also as potential pharmacological targets and therapeutic agents for neurological diseases. Molecules secreted by astrocytes play an important role in the pathophysiological processes that affect the astrocyte population. These molecules are involved in two key processes: (1) regulation of neural stem cells (NSCs) and their progeny in the adult neurogenic niche and (2) modulation of the integrity and function of the blood–brain barrier (BBB). Astrocytes provide structural and functional support for neurons by providing nutrients and neurotrophic factors and removing neurotransmitters and metabolic waste. This helps to maintain a homeostatic environment for neuronal function [31]. It is believed that astrocytes may also release gliotransmitters to modulate synaptic transmission [32,33]. Additionally, after a brain injury, astrocytes play a role in neuroinflammatory responses as they attempt to repair and/or remold the tissue.

Unlike other types of stem cells presented, GPCs do not replace neurons by integrating them into the neural network. Instead, these cells differentiate into oligodendrocytes, leading to the active myelination of axons, or into astrocytes, offering support and performing paracrine regulation of the nerve cells [34]. Despite the fact that embryonic GPCs show promise as potential therapeutic agents, they have limitations in their initial quantity and ability to reproduce. As a result, it is necessary to periodically obtain new GPCs from donor tissue [35].

### 2.3. Mesenchymal Stem Cells

MSCs are the most widely studied cell type in adult stem cell therapy, due to their abundance and extensive potential for differentiation. These cells can be administered intravenously, allowing them to cross the blood–brain barrier and evade the body’s immune response. In particular, extracellular vesicles derived from MSCs have been shown to exhibit donor properties with reduced immunogenicity, making them a promising option for use in stem cell therapy. However, there is still a small risk of tumor formation following treatment [36]. In 2019, Reza-Zaldivar et al. found that extracellular vesicles from mesenchymal stem cells (MSCs) can increase neuronal plasticity and improve cognitive function [37]. They injected Aβ1-42 amyloid aggregates into the dentate gyrus of mice and performed a novel object recognition test 14 and 28 days after vesicle administration. The study showed that the vesicles stimulated neurogenesis in the subventricular zone, leading to improved cognitive function in the mice. Administration of MSCs enhanced neurogenetic activity as well [38]. In an experiment to prevent the atrophy of the cholinergic system, the motor and cognitive functions of elderly mice improved after MSC transplantation. The conclusions were drawn based on the results of video tracking, water maze testing, and passive avoidance tests [39].

In 2020, Nakano et al. found that bone marrow-derived mesenchymal stem cells (BM-MSCs) can improve cognitive function in the AD model by increasing the expression of miR-146a in the hippocampus [40]. The cells were injected intracerebroventricularly into the choroid plexus of the lateral ventricles, where they released exosomes containing miR-146a into the cerebrospinal fluid. In vitro experiments showed that astrocytes took up the exosomal miR-146a, and its level increased. At the same time, Wei et al. investigated whether MSC-derived miR-223 regulates neuronal cell apoptosis by targeting PTEN and activating the PI3K/Akt pathway [41]. This could potentially be a treatment for AD, as it inhibits neuronal apoptosis.

Recently, clinical studies have also made significant progress, but they are primarily based on MSC therapy. In 2021, Kim et al. performed intracerebroventricular injections of human umbilical cord blood MSCs in patients with AD in a phase I clinical trial [42]. Nine patients with mild to moderate AD were enrolled and received low and high doses of MSCs, respectively. Within 36 h, all adverse effects had ceased, and AD symptoms were alleviated. MSC therapy reduces neuroinflammation by eliminating beta-amyloid, neurofibrillary tangles, and abnormal protein degradation. It promotes the restoration of the blood–brain barrier, regulates acetylcholine levels, improves cognitive functions, and helps autophagy in the brain [43,44].

MSCs have the ability to replace damaged nerve cells and have low immunogenicity. They also have a paracrine effect on their microenvironment. However, MSC therapy has some limitations. Studies have shown that the rate of neuronal differentiation is low, and it depends on the brain microenvironment of the recipient. The differentiation pathway can vary depending on the microenvironment, which plays a significant role in determining the outcome of MSC transplantation [45].

### 2.4. Embryonic Stem Cells

Due to the ethical and immunological limitations of using ESCs for therapeutic purposes, clinical applications of ESC-based treatments may not be promising. However, preclinical research on ESCs in animal models is ongoing [43]. The main ability of ESCs is to replace any type of cell, depending on the specific tissue environment in which they are introduced. ESCs were injected into a mouse with AD, leading to the formation of cholinergic neurons and improved synaptic connections, resulting in enhanced memory [25,46,47]. In addition, neurospheres derived from mouse ESCs have been reported to generate cholinergic neurons in the cerebral cortex of mice with basal nucleus lesions, which improved their working memory [48]. Motor neuron precursors derived from ESCs treated with Sonic hedgehog (SHH) and retinoic acid (RA) were found to differentiate into motor neurons after being transplanted into the basal forebrain, improving cognitive function in the affected areas of rat brains [46]. ESCs from both mice and humans were also differentiated into basal forebrain cholinergic neuron (BPCN) precursors and implanted into the brains of mice with AD. Two months after injection, the implanted progenitor cells predominantly differentiated into mature cholinergic neurons. Neuronal precursor therapy has been shown to relieve cognitive deficits in two strains of mice with AD (5xFAD and APP/PS1), within six months following transplantation [49,50]. In addition, human ESCs have been found to give rise to neural progenitor cells from the medial ganglionic eminence after treatment with SHH. These transplanted cells have differentiated into basal forebrain cholinergic neurons (BPCNs) and GABAergic interneurons, helping to negate learning and memory deficits in mice with damaged medial septal nuclei [51]. ESC therapy improved cognitive function in 5xFAD transgenic mice. Spontaneous alteration performance in the Y-maze was used to test the spatial working memory of the mice and contextual fear-conditioning testing was carried out [46]. Despite the obvious advantages of pluripotency, it is becoming one of the main problems of using ESCs. In many studies, researchers have observed the uncontrolled differentiation of these stem cells and the formation of teratomas [52,53].

### 2.5. Induced Pluripotent Stem Cells

The emergence of technology that allows somatic cells to be reprogrammed into pluripotent stem cells has made it possible to create a model that preserves the genetic identity of the donor. This offers an alternative to regenerative therapies. Induced pluripotent stem cells (iPSCs) can self-renew indefinitely in vitro and differentiate into various cell types, providing prospects for the modeling and treatment of AD in some patients [54]. At the genetic and cellular levels, numerous studies have been conducted on iPSC therapy for AD. In 2020, Butler et al. found the genetic relevance of microglia derived from human iPSCs for AD [55]. Using gene expression specific to these cells, it has been demonstrated that microglia cells derived from iPSCs are genetically linked to AD. Additionally, in 2020, Zhang et al. discovered that nerve cells derived from the iPSCs of AD patients exhibit varying susceptibilities to oxidative stress [56].

The response of nerve cells to oxidative stress is a significant mechanism of cognitive decline and aging. When exposed to hydrogen peroxide, the vitality and length of neuronal extensions in human neurons are significantly reduced. Because of the oxidative nature of neurons, there is potential for treatments for AD that focus on deoxidizing them. Due to the minimally invasive production of iPSCs without the use of any embryonic material [57,58], and the improved control of cell fate determination in vitro, iPSC differentiation into neurons can be triggered.

In a mouse model of AD, neural progenitors derived from iPSCs showed improvements in memory and reduced synaptic abnormalities. Researchers stereotactically injected murine mPSC neural precursors (iPSC-NPCs) into the hippocampus of mice, leading to an improvement in synaptic plasticity and a decrease in the number of glomerular and amyloid deposits [59]. Neural precursors of the cholinergic phenotype derived from iPSCs were transplanted into the hippocampus of a promoter-controlled amyloid precursor protein (PDAPP) transgenic mouse model of dementia [60]. After 45 days, it was found that the cells had survived and differentiated into cholinergic and GABAergic neurons in the host brain. This resulted in improved spatial memory in the mice [61]. Intranasal administration of the iPSC-derived NSC secretome led to memory restoration, reduced Aβ plaque accumulation, and increased neuronal proliferation [62].

In another study, iPSC derivatives, macrophage-like cells (iPSC-MCs), were developed to express neprilysin-2 (NEP2), a secreted protease with Aβ-degrading activity. iPSC-MK/NEP2 was injected into the hippocampus of a transgenic 5xFAD mouse [63,64]. Although the effects on cognitive function and neuronal damage were not studied, there was a significant reduction in Aβ levels in mice brains. This reduction in Aβ was not seen with transplantation of unmodified iPSC-MCs, demonstrating that NEP2 secretion rather than phagocytosis caused Aβ elimination. This study suggests a potential therapeutic benefit of NEP2-secreting iPSC-MCs for the treatment of AD. Human iPSCs cultured with SHH, RA, and Noggin differentiated into neural progenitors, which were shown to give rise to cholinergic and GABAergic neurons in the hippocampus of mice with AD and to promote the restoration of spatial memory [61]. Thus, significant efforts have been made in recent years to investigate stem cell-based AD replacement therapy.

In an experiment with a 5xFAD mouse line, which has five familial Alzheimer’s disease-associated mutations in App/PSEN1 genes, injections of iPSCs restored cognitive impairment. The “Y-maze” test was used to test the spatial working memory of mice and contextual fear-conditioning testing was also conducted [65]. It is worth noting that iPSCs require increased attention, especially during the reprogramming stage, due to the potential for a pathological phenotype [66]. Additionally, iPSCs may differentiate based on the type of donor cells and can inherit epigenetic features from the original cell type [67].

## 3. Mechanisms of Therapeutic Effect of Stem Cells in AD

The therapeutic potential of stem cells has been studied in different target brain regions of animals with AD. The studies focused on the hippocampus and basal forebrain, which are closely related to cognitive function. Once the benefits of NSCs, especially human NSCs, were evaluated in treating cognitive deficits in animals with AD, researchers began to investigate the possible mechanisms behind the action of the transplanted NSCs. These mechanisms are interpreted as cellular neuroprotection, leading to increased hippocampal synapse density and improved hippocampus-dependent cognition [25]. Interestingly, BDNF-mediated cognitive recovery does not affect the pathology of Aβ or tau in AD brains, indicating that BDNF acts through an amyloid-independent mechanism [25]. Hence, BPCN precursors corrected cognitive deficits in AD mice without altering total Aβ plaque levels. This has been partially demonstrated to be due to increased BDNF secretion [49,50,68]. BDNF and its active form, mBDNF, play an important role in the development of neurons, survival, and synaptic plasticity by activating the TrkB receptor and triggering various signaling pathways. These pathways include MAPK/ERK, PI3K, and PLCγ/PKC [69,70], which are all associated with the activation of the transcription factor CREB [71]. It, in turn, regulates the activity of genes that are required for synaptic plasticity. Other studies have suggested that NSC transplantation in mice reduces tau phosphorylation through Trk-dependent Akt/GSK3β signaling, leading to a decrease in Aβ production. This is mediated by Akt/GSK3β signaling and is assisted by the down-regulation of inflammatory mediators due to the deactivation of microglia. This deactivation can be facilitated by cell-to-cell contact or the secretion of anti-inflammatory factors from NSCs [30,72].

The neuroprotective effects of transplanted NSCs in preventing neuronal degeneration or atrophy, as well as synapse loss, were similar to those observed when BDNF was injected directly into the brains of patients with AD [73,74]. In addition, the transplanted BDNF precursors secreted acetylcholine and produced acetylcholinesterase in the basal forebrain regions of mice with AD, which was essential for the restoration of cognitive functions [49,50]. These findings suggest that neurons derived from NSCs exhibit similar functions to those in vivo with respect to the metabolism of acetylcholine in the brain in AD. Therefore, the neuroprotective effects of transplanted NSCs or neuronal precursors seem to be largely due to the secretion of neurotrophic factors or neurotransmitters, which may contribute to repairing brain damage and correcting cognitive deficits in patients with AD. In addition to BDNF, there are other factors that contribute to the protection of neurons from oxidative stress and neuroinflammation, such as NGF, NT-3, and Nrf2, as well as a decrease in iNOS and COX-2, and an increase in the expression of *SYN1* and *Sirt1* [75,76].

## 4. Stem Cell Replacement Therapy

Structural integrity and adequate synaptic activity are essential for normal brain function. After further studying transplanted NSCs and their derivatives, it has become evident that these cells can functionally replace damaged or degenerated neurons, in addition to providing neuroprotection for the AD brain. This demonstrates their ability to undergo terminal neuronal differentiation and survive over the long term. This confirms that NSCs can tolerate the conditions of the pathological brain environment in AD [49,51,61,77]. In addition, it has been found that differentiated progenitor cells can trigger action potentials and spontaneous postsynaptic currents in mice with atopic dermatitis damage, indicating that neurons derived from human NSCs have membrane properties similar to those of mature neurons [51].

Recently, several studies have systematically characterized the survival, proliferation, differentiation, migration, and integration of neurons derived from NSCs in mice with AD. These studies aim to assess the potential of NSC-derived neurons to replace lost brain cells and degenerated synapses in AD patients [49,68]. The migration of NSCs after transplantation is a critical factor that can greatly influence their therapeutic effectiveness. NSCs have the ability to migrate toward areas of neurodegenerative damage [16]. Their successful migration to these sites can enhance their ability to repair damaged tissue and provide neuroprotection. The timing and extent of NSC migration can affect therapeutic outcomes. Early migration may provide immediate benefits, while prolonged presence can lead to long-term advantages [30]. The surrounding microenvironment, including the presence of inflammatory factors or extracellular matrix components, can affect NSC migration and differentiation. Improving these conditions could enhance therapeutic efficacy [78].

It has been confirmed that transplanted BPCN precursors facilitate neuron maturation and exhibit migration patterns similar to native neurons in the basal nucleus of AD mice. Complex dendritic branches and long axons were found in transplanted BPCNs, and synaptic structures were commonly observed between exogenous and endogenous neurons using electron microscopy. In addition, most of the transplanted BPCNs showed excitatory and inhibitory synaptic activity, suggesting that they could functionally integrate into the cholinergic system in the basal forebrains of AD mice. It was also shown that human NSCs differentiated into glutamatergic neurons after one month of transplantation into the AD mice hippocampus. Glutamatergic neuronal grafts showed long-term survival of up to 12 months and remained healthy without Aβ plaque invasion or activated microglia. Optogenetic analyses confirmed that exogenously introduced neurons form synaptic connections with endogenous hippocampal neurons and exhibit appropriate postsynaptic activity after being transplanted into the host brain. This increases synaptic transmission and enhances local neural circuits in the brain, resulting in increased long-term potentiation and plasticity of hippocampal cells. Additionally, it has been shown to mitigate cognitive deficits associated with AD. These findings strongly suggest that human NSCs can functionally integrate into neural networks, replace damaged neurons, and strengthen impaired synaptic circuits in the brains of patients with AD.

## 5. Paracrine Action of Stem Cells

Alzheimer’s disease is associated with impaired secretion and signaling of various neurotrophins, growth factors, and cytokines. The pathological processes of AD result in a decrease in the basal levels of BDNF [79], IGF-1 [80], VEGF [81], and other signaling molecules. Paracrine effects play a key role in cell therapy, providing interactions between cells that can enhance the immune response or suppress inflammation, as well as restore neural functions and improve the microenvironment, creating conditions for regeneration and slowing the progression of the disease [82]. For example, the administration of MSCs in mice with Alzheimer’s disease led to increased levels of BDNF and IL-10, which correlated with improved cognitive function and decreased neuroinflammation [83]. At the same time, MSCs can secrete factors such as IL-6 and VEGF, which promote angiogenesis and improve cerebral blood flow [84]. Other work highlights the role of the MSC-secreted neurotrophic factor GDNF in maintaining neuron survival and reducing neuroinflammation [85]. At the same time, stimulation of cell survival by MSCs is associated with activation of signaling pathways such as PI3K/AKT and ERK, which leads to increased cell viability [86,87]. Another type of cell used in AD cell therapy is NSCs. They can secrete many factors, including BDNF, VEGF, and IGF-1, which support neuronal survival, promote neuroplasticity, and improve cerebral blood supply [22]. VEGF-A shows neuroprotective effects against oxidative stress, the neurotoxicity caused by Aβ, and the destabilization of the cytoskeleton due to hyperphosphorylation of tau [88]. Previous studies have also shown that administering IGF-1 to mice with AD leads to the activation of the GABA system, mTOR autophagy signaling in the hippocampus, and an improvement in synaptic activity as well as a reduction in neuroinflammation [89]. NSC-based therapy with intracerebral administration led to an improvement in memory and cognitive functions by secreting a significant amount of BDNF [25]. Studies have also shown that an increase in BDNF levels in Alzheimer’s disease leads to a reduction in the production of toxic Aβ [90] through alpha-secretase processing of APP, as well as inhibition of tau phosphorylation. In addition, NSCs secrete the IL-10 factor, which suppresses inflammatory processes, reducing the level of activated microglia [21]. Other studies using NSCs show that their paracrine factors, such as NRG-1, can enhance neuroplasticity and contribute to the repair of damaged synapses [91]. Thus, paracrine regulation of stem cells can not only promote neuroregeneration but also modify inflammatory responses in the brain.

## 6. Exosomes as Therapeutic Agents

During cell therapy, secreted exosomes play an important role in intercellular communication and restoration of neural function [92]. Exosomes contain proteins, lipids, microRNAs, and other molecules that can affect the functions of neighboring cells and promote regeneration [93]. microRNAs in exosomes can modulate gene expression in target cells, promoting neuronal survival, reducing oxidative stress, and improving neuroplasticity [94]. For example, microRNAs contained in MSC exosomes, such as miR-21 and miR-29b, can modulate cellular signaling pathways associated with neuron survival and neuroplasticity [95,96]. Exosomes may contain molecules that reduce inflammatory responses, which is especially important in neurodegenerative diseases such as Alzheimer’s disease. miR-146a has been shown to regulate the inflammatory response by inhibiting signals leading to the activation of microglia and the release of pro-inflammatory cytokines [97]. At the same time, the introduction of exosomes themselves into Alzheimer’s disease models led to an improvement in cognitive functions, which was associated with an increase in BDNF levels in the brain and a decrease in oxidative stress [98]. A similar effect was revealed with the introduction of NSC exosomes. They protected neurons from damage associated with oxidative stress and inflammation, and improved cellular metabolism [99]. At the same time, NSC exosomes stimulated neurogenesis (formation of new neurons) and synaptogenesis (formation of synapses), which is especially important for the treatment of neurodegenerative diseases [23]. Various studies have shown that MSC exosomes (MSC-EV) enhance angiogenesis and promote functional recovery via neurovascular remodeling [100,101], while NSC exosomes (NSC-EV) inhibit neuroinflammation and microglial activation [23,102]. Gao et al. have demonstrated that NSC-EV has a better effect on cognitive tests, such as the water maze test and fear-conditioning test. The exosomes show a more pronounced inhibition of Aβ1-42 accumulation and Tau phosphorylation, as well as better therapeutic effects on restoring dendrite length and the density of dendritic spines compared to MSC-EV in the mouse AD model [103].

## 7. Mitochondrial Transfer by Tunneling Nanotubes

The transfer of mitochondria through tunneling nanotubes (TNT) is an interesting mechanism that may have a significant impact on the pathogenesis of AD [104]. Mitochondrial dysfunction plays a significant role in the pathogenesis of AD. This condition is linked to a range of metabolic and cellular disorders that can aggravate the development of neurodegeneration. Mitochondrial dysfunction can lead to increased formation of reactive oxygen species (ROS) [105]. Excessive levels of ROS can cause damage to cellular structures, including lipids, proteins, and DNA. This can contribute to neurodegenerative conditions. In AD, the process of mitophagy, which is the removal of damaged mitochondria, is also disrupted. This leads to the accumulation of damaged mitochondria and a deterioration of cellular metabolism [106]. Ultimately, this leads to energy deficiency in neurons, which is critical for their normal functioning, as well as to the activation of apoptosis, which causes the death of neurons [107,108]. Mitochondrial transfer by stem cells via TNT is considered a potential corrective action for mitochondrial deficiency in AD [109]. Stem cells can transfer their mitochondria to damaged neurons, which may help restore mitochondrial function [110]. By restoring or increasing the number of functional mitochondria in neurons, energy production may be improved, which is crucial for normal functioning. Stem cell-derived mitochondria may have higher antioxidant activity, helping to reduce free radical levels and thereby reducing oxidative stress in neurons. This mitochondrial transfer may also activate defense mechanisms within neurons, promoting their survival and reducing apoptosis [111].

Studies show that MSCs can use TNT to transfer healthy mitochondria to neighboring neurons. In some animal experiments, it has been shown that MSC injections can lead to increased neuron survival and improved cognitive function due to mitochondrial transfer [112]. NSCs can also form TNT, which binds them to neighboring neurons, allowing for the transfer of mitochondria. At the same time, neurons that received mitochondria from NSCs showed significantly higher ATP levels and reduced oxidative stress, which indicated an improvement in their metabolic activity, which led to improved cognitive functions in the Alzheimer’s disease model [113].

## 8. Challenges and Promises of Cell Therapy for Alzheimer’s Disease

Cell therapy is a promising approach to treating neurodegenerative diseases, aiming to trigger regenerative processes and restore brain functions. However, conducting clinical trials in this field is associated with several technical challenges, in order to accelerate the development of effective treatments for Alzheimer’s disease. One of the main concerns regarding this therapy is its safety. MSCs have been identified as a promising source for clinical trials, due to their high biosafety and their ability to produce neurotrophic and pro-angiogenic factors [114]. Another challenge faced by researchers is determining the best time for stem cell transplantation. Since AD is a chronic and progressive condition, by the time symptoms or signs of the disease appear, the patient’s brain may have already suffered significant damage, making treatment less effective [82]. The choice of how to administer stem cells is another limiting factor. Although intracerebral administration allows for targeted delivery of the transplant without needing to overcome the blood–brain barrier, it often requires serious surgical intervention [115]. On the other hand, intravenous administration of stem cells is a minimally invasive and simple method of delivery. However, this method is not very effective in treating AD as the cells can be captured by the liver, spleen, and lungs before reaching target tissues [116]. Cell therapy raises ethical concerns such as the source of stem cells and ensuring adequate awareness among donors and patients. Human ESCs are derived from a blastocyst, which is a 5–7-day-old embryo. In many countries, research on human embryos is restricted. Due to the concerns about creating a market for producing embryos and the risk of women being exploited, it is rather cumbersome to establish the transfer of embryos. Nevertheless, stem cells donated on a voluntary basis are currently the most accepted way to obtain ESCs. It follows that in order to use both “unnecessary to parents” blastocysts and to receive donor ones, it is necessary to create strict international regulations according to which ESCs will be recorded and all relevant documentation will be kept. It is important to determine the appropriate amount of information to provide to patients undergoing stem cell treatment. It is not ethical to mislead seriously ill patients by offering unproven methods of treatment.

There are currently several clinical trials underway on stem cell therapy for AD. Intracerebroventricular administration of human umbilical cord blood-derived mesenchymal stem cells has proven to be feasible, relatively safe, and well tolerated with a reduction in levels of Aβ42 and phosphorylated Tau [42]. Clinical trials have also been conducted using NSCs administered intranasally, although the results of these trials have not been released yet [117,118]. Despite the promising results seen in preclinical studies involving ESCs and iPSCs, in contrast, clinical trials for AD using these cells have not yet begun. The use of ESCs in the treatment of AD raises ethical concerns due to their origin and the potential for immunological rejection. Similarly, the use of iPSCs also faces challenges attributed to immunological reactions and the risk of tumor development [119]. 

Other possible side effects include acute fever on the first day after administration and an increase in hippocampal volume. These side effects are associated with the delivery method using intracranial probes, which may lead to infections [42,120]. Additionally, the Food and Drug Administration (FDA) has issued guidelines for the use of Lecanemab that reflect its low efficacy and that recommend this drug only for patients in the early stages of AD. In addition, people with autoimmune diseases, depression, and those who have suffered stroke or ischemic injuries were excluded from the second and third stages of clinical trials of Lecanemab [121]. This suggests that this drug should not be used widely and requires further research. The use of MSCs and iPSCs is associated with a lower risk of inflammation and rejection. Many research groups believe that combining cell therapy with anti-beta-amyloid antibody treatments may yield more positive outcomes [122].

Recently, the FDA granted Fast Track designation status to Lomecel-B, a cellular preparation developed by Longeveron [123]. This drug consists of MSCs derived from the bone marrow of healthy donors, and it has shown promising results in clinical trials, improving cognitive functions and slowing down the rate of brain volume loss in AD patients [120]. Based on data from the IIa phase of clinical trials, brain MRI for Amyloid-Related Imaging Abnormalities (ARIAs) was not observed in participants receiving Lomecel-B treatment [124]. In contrast, Lecanemab has been linked to ARIAs, which can manifest as temporary swelling of the brain and may also cause small hemorrhages. This condition may cause headaches, dizziness, vision changes, and nausea in some patients. However, considering that ARIAs have not been observed for Lomecel-B-treated patients and no adverse effects have been reported, there is hope that MSC-based therapy could become available to more patients in the future. Overall, there seems to be a trend towards cellular-based therapies, and it is possible that more such drugs may become available on the market soon.

## 9. Conclusions

Alzheimer’s disease is a progressive brain disorder affecting memory and cognition, characterized by cholinergic neuron loss, protein deposition, and brain cell dysfunction. Current therapies have limited efficacy due to incomplete understanding of underlying mechanisms and the need for long-term treatment. The exploration of cell therapy for AD represents a promising frontier in the quest for effective treatments. As our understanding of neurodegenerative mechanisms deepens, stem cell therapy is emerging as a compelling option that not only addresses cognitive decline but also facilitates neuronal regeneration and reduces neuroinflammation. Cell therapy poses a promising frontier in the treatment of Alzheimer’s disease, particularly through the application of various stem cell types (Figure 1). The potential of neural stem cells (NSCs) to restore cognitive function, reduce neuroinflammation, and target the underlying pathology of AD highlights their importance in regenerative medicine, demonstrating significant improvements in cognitive functions and a reduction in pathological markers in animal models of AD (Table 1).

Moreover, the insights gained from studies utilizing mesenchymal stem cells (MSCs), embryonic stem cells (ESCs), and induced pluripotent stem cells (iPSCs) further emphasize the versatility of stem cell applications in combating neurodegeneration. Continued exploration of these cellular therapies may lead to breakthroughs that not only enhance cognitive function but also improve patients’ quality of life. While challenges remain concerning the long-term efficacy and safety of these therapies, as well as ethical and legal issues that were not touched on in this review, the initial outcomes from recent research are encouraging. Simplified methods for obtaining safe autologous iPSCs could be a solution to the problems faced by researchers, as a step towards personalized medicine, or by rejecting stem cell therapy in favor of paracrine factors produced by these cells. Currently, there are multiple developments that are underway for stem cell therapies to improve targeted delivery or increase their survival rates. However, there are still no clear protocols available that would ensure predictable therapeutic outcomes.

The knowledge about the mechanisms of action of stem cells is constantly becoming more complex and extensive. The hypothesis of the replacement action of stem cells has been changed by an understanding of the diverse properties of stem cells. Among them are the regulation of immunity, neuroplasticity, inflammatory reactions, and others through paracrine effects via released soluble factors or extracellular vesicles, as well as contact involving the formation of tunneling nanotubes. Future studies should focus on refining delivery mechanisms, optimizing cell types, and elucidating the underlying mechanisms by which stem cells exert their beneficial effects. By harnessing the regenerative abilities of stem cells, we might not only slow the progression of Alzheimer’s disease but also promote recovery and enhance the quality of life for those affected. In addition, these types of cell therapies could be potentially extended to other neurodegenerative disorders such as Parkinson’s disease, amyotrophic lateral sclerosis, Huntington’s disease, and others. Continued collaborative research in regenerative medicine will be crucial for translating these findings into viable clinical therapies capable of combating devastating conditions. The ongoing efforts to unravel the complexities of Alzheimer’s disease and the role of stem cells therein may eventually revolutionize treatment approaches and offer hope to millions affected by this debilitating condition. As we move forward, collaboration across disciplines and a commitment to further studies will be essential in unlocking the full potential of cell therapy in the fight against neurodegenerative disorders. Overall, stem cell therapy holds promise for treating Alzheimer’s disease, with ongoing research focused on understanding mechanisms and optimizing treatment strategies.

## Figures and Tables

**Figure 1 ijms-25-12378-f001:**
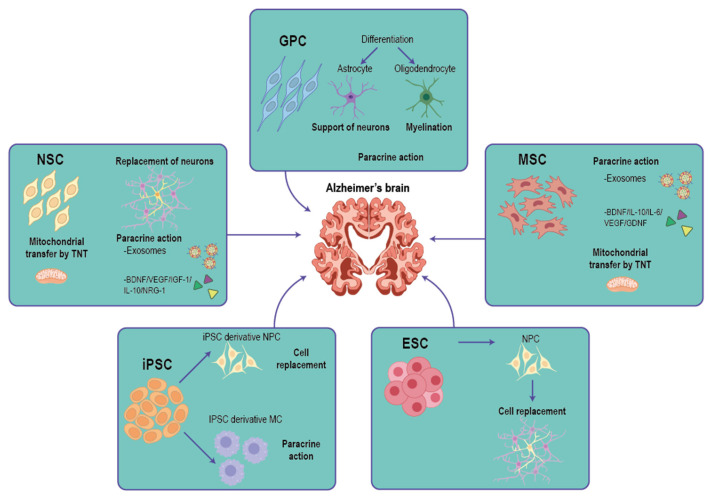
Stem cell strategies for Alzheimer’s disease and proposed mechanisms of stem cell therapy. Mesenchymal stem cells (MSCs), neural stem cells (NSCs), glial progenitor cells (GPCs), neural progenitor cells (NPCs), embryonic stem cells (ESCs), and induced pluripotent stem cells (iPSCs) participate in various types of cell processing affecting cell replacement and are involved in paracrine action via secreted factors and exosomes, as well as the transfer of mitochondria through tunneling nanotube (TNT) pathway.

**Table 1 ijms-25-12378-t001:** Therapeutic effects of stem cells in Alzheimer’s disease.

Cell Types	Major Mechanism or Target	Reference
Mesenchymal stem cells	Increase in neuroplasticity, stimulation of neurogenesis	[37,38]
	Enhancement of microRNA expression in the hippocampus	[40,95,96,97]
	Inhibition of neuronal apoptosis through activation of the PI3K/Akt signaling pathway	[41,86,87]
	Reduction in neurofibrillary tangles and β-amyloid	[42,44,87]
Neural stem cells	Improvement of cognitive functions and regeneration of nerve tissue	[21,23,28,29]
	Reduction in β-amyloid accumulation	[23,30]
	Reduction in the production of pro-inflammatory cytokines	[24,28,30]
	BDNF-mediated neuroprotection and increased hippocampal synapse density	[25,98]
Glial stem cells	Regulation of NSC	[31]
	Reduction in neuroinflammation	[33]
	Modulation of synaptic transmission by released glial transmitters	[32]
Embryonic stem cells	Generation of cholinergic neurons and enhancement of synaptic formation	[25,46,47]
	Regeneration of damaged brain regions, improved memory function	[48,51]
Induced pluripotent stem cells	Improvement of memory and synaptic function	[59,62]
	Reduction in neurofibrillary tangles and β-amyloid plaque formation	[62,63]

## Data Availability

The data presented in this study are available from the corresponding author upon reasonable request.

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
