# Peer review of "Proposed Mechanisms of Cell Therapy for Alzheimer’s Disease"

_ijms, 2024, doi:10.3390/ijms252212378_

Round 1

Reviewer 1 Report

Comments and Suggestions for Authors

ijms-3291949

Proposed Mechanisms of Cell Therapy for Alzheimer’s Disease

The manuscript by Belousova et al. summarized preclinical and clinical studies on the use of novel types of stem cell treatment approaches for Alzheimer’s Disease. The manuscript was well constructed and might contribute to the field. However, it may need improvements, considering the comments below.

1. The Introduction should be expanded to clarify: (i) why stem cell treatment approaches for Alzheimer’s Disease are critical, (ii) how they have been developed in recent years, (iii) what is the motivation to conduct this review, and (iv) what are the novelty and contribution of this review.

2. Sections 4, 5, 6, and 7 are relatively superficial. Please work on expanding them and adding more references, comparisons, and discussion.

3. Please discuss the challenges and future prospects of cell therapy for Alzheimer’s Disease. Will it be soon available for clinical practice?

4. Please include some figures to illustrate the information summarized in section 4, 5, and 6.s

Author Response

Dear Reviewer,

We are grateful for your time and effort spent reviewing our manuscript. We highly appreciate your constructive feedback and thoughtful comments.

We revised the manuscript according to your comments.

  1. The Introduction should be expanded to clarify: (i) why stem cell treatment approaches for Alzheimer’s Disease are critical, (ii) how they have been developed in recent years, (iii) what is the motivation to conduct this review, and (iv) what are the novelty and contribution of this review.

The Introduction was expanded with information about different methods of cellular therapy and the motivation of the review.

  1. Sections 4, 5, 6, and 7 are relatively superficial. Please work on expanding them and adding more references, comparisons, and discussion.

Sections 4, 5, 6, 7 were supplemented with information about the disadvantages of using a particular type of cells, the mechanisms of action of some paracrine factors in Alzheimer's disease, the effects of MSC and NSC exosomes and their differences.

  1. Please discuss the challenges and future prospects of cell therapy for Alzheimer’s Disease. Will it be soon available for clinical practice?

We added a new section 8 that is devoted to the challenges and promises of cell therapy for Alzheimer's disease. It examines the problems associated with applying cellular therapy in clinical practice, as well as some clinical trials of certain types of cells in this disease.

  1. Please include some figures to illustrate the information summarized in section 4, 5, and 6.

Thank you for the remark. A diagram was added to the manuscript, which summarizes the information provided in the review about the main mechanisms of action of various types of stem cells in Alzheimer's therapy.

Reviewer 2 Report

Comments and Suggestions for Authors

The manuscript comprehensively reviews the mechanisms underlying various stem cell therapies for Alzheimer’s disease. It examines each stem cell type’s neuroprotective, immunomodulatory, and regenerative potentials, offering a balanced assessment of experimental outcomes and therapeutic mechanisms. The report's insights into how stem cells act through paracrine signaling, exosome release, and mitochondrial transfer bring a nuanced perspective to ongoing AD therapeutic research. However, several improvements are suggested below to enhance scientific clarity, precision, and engagement with recent literature:

1.      The current manuscript mentions each stem cell type and its primary action, but it would benefit from a more precise explanation of how these mechanisms differ. For example, neural stem cells (NSCs) can replace damaged neurons and form synaptic connections, whereas mesenchymal stem cells (MSCs) often function primarily through paracrine signaling. Expanding on these distinctions clarifies why specific stem cell types may be more suitable for certain therapeutic goals in Alzheimer’s disease (AD) therapy. This would add depth and specificity to the review, improving readers' understanding of why multiple stem cell types are considered for AD.

2.      The manuscript should clearly distinguish between cell replacement therapy and cell support therapy in the introduction (Lines 11–14). Stem cells can serve two main functions: replacing damaged neurons or supporting existing neural structures through signaling and neuroprotective functions. These approaches target different aspects of AD pathophysiology, which can help readers understand the broader therapeutic strategies. This clarification could also set a foundation for later sections, where different stem cell types align with either replacement or support functions.

3.      Ethical issues associated with embryonic stem cells (ESCs) and induced pluripotent stem cells (iPSCs) are fundamental to their clinical applications (Line 145–146). While the manuscript briefly touches on the limitations of ESCs due to ethical concerns, it would benefit from a more nuanced discussion of these concerns. For instance, ESCs face ethical and immunological challenges, and iPSCs, though ethically viable, can have genetic instability. By acknowledging these complexities, the manuscript would provide a balanced view of potential clinical obstacles, aligning with the audience’s likely awareness of ethical considerations in scientific research.

4.      Paracrine signaling by stem cells plays a significant role in their neuroprotective effects, mainly through secreted factors such as BDNF, VEGF, and IGF-1. These molecules contribute differently to synaptic repair, neurogenesis, and neuroprotection (Lines 264–281). Expanding on how these factors act individually and in concert would enhance this section, allowing readers to understand the therapeutic impact of paracrine signaling and appreciate its complexity. A deeper discussion of this signaling would also provide insight into how these processes potentially improve cognitive function in AD models.

5.      In the current manuscript, several references are cited without context for their contributions. For example, the authors should elaborate on why certain studies, such as those involving microglial functions, were chosen to support specific mechanisms (Lines 20–25).

6.      The manuscript discusses exosomes as therapeutic agents but could delve further into how exosomes from different stem cell types vary in function (Lines 287–303). For example, NSC exosomes may have distinct neurogenic and anti-inflammatory effects compared to MSC exosomes. Detailing these functional differences would make the review more comprehensive and clarify why NSC and MSC exosomes are considered. This would also help emphasize stem cell-derived exosome therapy's versatility and tailored potential.

7.      The section on mitochondrial transfer (Lines 306–319) would benefit from added background on mitochondrial dysfunction in AD, which would help explain the therapeutic importance of tunneling nanotubes (TNT). Since mitochondrial damage contributes to AD progression, including this information would provide context for readers less familiar with cellular bioenergetics. This would establish a logical flow where mitochondrial transfer by stem cells through TNTs is introduced as a corrective action for mitochondrial deficits in AD.

8.      Adding studies demonstrating clear outcomes on cognitive improvement in AD models would reinforce the connection between the mechanisms described and the observed therapeutic efficacy (Line 71–74). For instance, citing specific memory test results in animal models after NSC therapy could strengthen the link between cellular mechanisms (e.g., BDNF release) and cognitive benefits. Including such studies would provide concrete evidence, enhancing the manuscript’s impact by linking therapeutic mechanisms to tangible outcomes.

9.      A section comparing the efficacy, safety profiles, and limitations of NSCs, MSCs, and iPSCs would be beneficial, helping readers grasp the strengths and weaknesses of each approach (Line 25). This comparative analysis would allow readers to see the “bigger picture” of stem cell therapies for AD and make it easier for them to assess which stem cell types might offer the most promise, especially considering the balance of risks and benefits.

10.   Including current clinical applications or trials would ground the review in practical terms (Line 142–143). For example, mentioning specific trials for MSC or iPSC therapies would contextualize the experimental findings, showing readers that these therapies are already under investigation for potential real-world applications. Additionally, discussing the logistical challenges of delivering stem cells to the brain could underscore the need for further innovation in clinical applications.

11.   How NSCs migrate post-transplantation can significantly impact their therapeutic efficacy. Discussing whether NSC migration is directed or random (Line 243–245) will help readers understand if transplanted cells localize to AD-affected areas effectively.

12.   The manuscript could elaborate on BDNF’s role in AD pathology, especially its ability to improve cognitive function independently of amyloid-beta levels (Line 215–220). A deeper exploration into this amyloid-independent mechanism could present BDNF as a therapeutic target, focusing on cognitive restoration even if amyloid reduction proves difficult.

13.   The conclusion currently calls for further research but could benefit from specific suggestions, such as refining delivery systems or bioengineering cells for better survival (Line 351–359). Proposing actionable future steps would guide researchers in exploring innovative ways to overcome current limitations in stem cell therapy for AD. Adding targeted recommendations would also position the manuscript as an influential source for shaping future research directions.

14.   A flowchart summarizing each stem cell type’s mechanisms (e.g., cell replacement, paracrine effects, exosome action, TNT mitochondrial transfer) would aid comprehension (Line 322). Such visual aids are helpful for complex reviews, allowing readers to quickly understand the scope of therapeutic actions associated with each cell type. Including a summary chart or figure would also enhance the review’s appeal as a comprehensive reference, especially for researchers or students visualizing mechanisms at a glance.

Author Response

Dear Reviewer,

Thank you very much for your careful reading of our manuscript and for providing such constructive feedback. We truly appreciate your time and effort in reviewing our work.

We added new materials according to your comments to improve the quality of our manuscript.

  1. The current manuscript mentions each stem cell type and its primary action, but it would benefit from a more precise explanation of how these mechanisms differ. For example, neural stem cells (NSCs) can replace damaged neurons and form synaptic connections, whereas mesenchymal stem cells (MSCs) often function primarily through paracrine signaling. Expanding on these distinctions clarifies why specific stem cell types may be more suitable for certain therapeutic goals in Alzheimer’s disease (AD) therapy. This would add depth and specificity to the review, improving readers' understanding of why multiple stem cell types are considered for AD.

Thank you for the comment. We have emphasized the main types of stem cell effects on nerve tissues in the manuscript text in the following Sections:

2.1. «As neural stem cells (NSCs) play a significant role in brain homeostasis and repair, they have been shown to have potential for the treatment of AD in the early stages, replacing neurons and astrocytes and exerting various paracrine effects [16]»

2.2. «Unlike other types of stem cells presented, GPCs do not replace neurons by embedding themselves into the neural network. Instead, GPCs differentiate into oligodendrocytes, leading to active myelination of axons, or into astrocytes, performing support and paracrine regulation of nerve cells [https://doi.org/10.1038/nm974]. »

2.3. «MSCs have low immunogenicity, ability to replace nerve cells and paracrine influence on its microenvironment.»

2.4. «The main ability of ESCs is to replace any type of cell, depending on the tissue environment in which they have been introduced.»

  1. The manuscript should clearly distinguish between cell replacement therapy and cell support therapy in the introduction (Lines 11–14). Stem cells can serve two main functions: replacing damaged neurons or supporting existing neural structures through signaling and neuroprotective functions. These approaches target different aspects of AD pathophysiology, which can help readers understand the broader therapeutic strategies. This clarification could also set a foundation for later sections, where different stem cell types align with either replacement or support functions.

The introduction was supplemented with information about different methods of cellular therapy. Moreover, the introduction was expanded with the motivation for the review.

  1. Ethical issues associated with embryonic stem cells (ESCs) and induced pluripotent stem cells (iPSCs) are fundamental to their clinical applications (Line 145–146). While the manuscript briefly touches on the limitations of ESCs due to ethical concerns, it would benefit from a more nuanced discussion of these concerns. For instance, ESCs face ethical and immunological challenges, and iPSCs, though ethically viable, can have genetic instability. By acknowledging these complexities, the manuscript would provide a balanced view of potential clinical obstacles, aligning with the audience’s likely awareness of ethical considerations in scientific research.

In section 2. Stem Cells in AD Therapy, new information on the disadvantages of using a particular type of cells as therapy were added for each type of cell.

2.1. «One of the main disadvantages of the NSC therapy method is poor viability. In many cases, losses can be as much as 90%. Therefore, NSC replacement requires additional conditions to maintain stem cell survival [doi: 10.1021/bc400005m, doi: 10.1016/j.expneurol.2013.04.001].»

2.2. «Despite the fact that embryonic GPC look promising as potential therapeutic vectors, they have limitations in their initial number and ability to reproduce. Because of this, it is re-quired to periodically receive new GPCs from donor tissues [DOI: 10.1126/science.1218071].»

2.3. «At the same time, MSC therapy is not flawless. Studies have noted a low rate of neuronal differentiation, and moreover, it depends on the microenvironment of the host brain. The differentiation pathway may differ by the microenvironment of one organism to another, which plays an important role in determining the fate of transplanted MSC [doi.org/10.1046/j.1440-1789.2003.00496.x].»

2.4. «Despite the obvious advantages of pluripotency, it is becoming one of the main problems of using ESC. In many studies, researchers have noted the uncontrolled differentiation of this type of stem cells and the formation of teratomas [doi: 10.1038/nbt726 , doi: 10.1016/S0002-9440(10)62488-1].»

2.5. «It is worth noting that iPSC requires increased attention, especially at the reprogramming stage due to the possible pathological phenotype [doi: 10.1186/s12864-015-1262-5]. iPSCs also tend to differentiate according to the type of donor cells and may inherit epigenetic features from the original ones [https://doi.org/10.1007/978-4-431-55966-5_1].»

  1. Paracrine signaling by stem cells plays a significant role in their neuroprotective effects, mainly through secreted factors such as BDNF, VEGF, and IGF-1. These molecules contribute differently to synaptic repair, neurogenesis, and neuroprotection (Lines 264–281). Expanding on how these factors act individually and in concert would enhance this section, allowing readers to understand the therapeutic impact of paracrine signaling and appreciate its complexity. A deeper discussion of this signaling would also provide insight into how these processes potentially improve cognitive function in AD models.
    Section 5 was expanded with additional information on the mechanisms of action of BDNF, VEGF and IGF-1 in Alzheimer's disease.

«Alzheimer's disease is associated with impaired secretion and signaling of various neurotrophins, growth factors and cytokines. Pathological processes of AD lead to a decrease in the basal levels of BDNF [https://doi.org/10.1016/j.neurobiolaging.2015.04.014], IGF-1 [https://doi.org/10.1006/nbdi.2000.0311], VEGF [https://doi.org/10.1016/j.neulet.2022.136799] and other signaling molecules.»

«VEGF-A show neuroprotective effects against oxidative stress, neurotoxicity caused by Aβ, and destabilization of the cytoskeleton due to hyperphosphorylation of tau [https://doi.org/10.3389/fnmol.2023.1181626]. As previous studies have shown, administering IGF-1 to mice with AD led to the activation of the GABA system and mTOR autophagy signaling in the hippocampus, as well as to an improvement in synaptic activity and a reduction in neuroinflammation [https://doi.org/10.3390/ijms25052567].»

«Studies also have shown that an increase in BDNF levels in Alzheimer's disease leads to a reduction in the production of toxic Aβ [https://doi.org/10.1111/jnc.14034] through alpha-secretase processing of APP, as well as inhibition of tau phosphorylation [https://doi.org/10.1016/j.neuropharm.2021.108737].»

  1. In the current manuscript, several references are cited without context for their contributions. For example, the authors should elaborate on why certain studies, such as those involving microglial functions, were chosen to support specific mechanisms (Lines 20–25).

Thank you for the comment. We have added information in introduction section about the importance of microglia in the progression of Alzheimer's disease and how cell therapy can influence on this target.

  1. The manuscript discusses exosomes as therapeutic agents but could delve further into how exosomes from different stem cell types vary in function (Lines 287–303). For example, NSC exosomes may have distinct neurogenic and anti-inflammatory effects compared to MSC exosomes. Detailing these functional differences would make the review more comprehensive and clarify why NSC and MSC exosomes are considered. This would also help emphasize stem cell-derived exosome therapy's versatility and tailored potential.

Section 6 was supplemented with new information on the effects of MSC and NSC exosomes.

«Various studies have shown that MSC exosomes enhance angiogenesis, promote func-tional recovery and neurovascular remodeling [10.1186/s13287-020-01834-0; https://doi.org/10.1038/jcbfm.2013.152], while NSC exosomes often inhibit neuroinflammation and microglia activation [DOI: 10.1186/s13195-021-00791-x; DOI: 10.1111/jnc.15001]. Gao et al.'s latest research has demonstrated that NSC-EV have a better effect in cognitive tests (water maze test, fear conditioning test), show a more pronounced inhibitory effect on Aβ1-42 accumulation and Tau phosphorylation, and have better therapeutic effects on restoring dendrite length and density of dendritic spines compared to MSC-EV in the mouse AD model [s41392-023-01436-1].»

  1. The section on mitochondrial transfer (Lines 306–319) would benefit from added background on mitochondrial dysfunction in AD, which would help explain the therapeutic importance of tunneling nanotubes (TNT). Since mitochondrial damage contributes to AD progression, including this information would provide context for readers less familiar with cellular bioenergetics. This would establish a logical flow where mitochondrial transfer by stem cells through TNTs is introduced as a corrective action for mitochondrial deficits in AD.

Thank you for comment. We added information about mitochondrial dysfunction in AD, and wrote how the transfer of mitochondria through TNT can affected on AD.

8. Adding studies demonstrating clear outcomes on cognitive improvement in AD models would reinforce the connection between the mechanisms described and the observed therapeutic efficacy (Line 71–74). For instance, citing specific memory test results in animal models after NSC therapy could strengthen the link between cellular mechanisms (e.g., BDNF release) and cognitive benefits. Including such studies would provide concrete evidence, enhancing the manuscript’s impact by linking therapeutic mechanisms to tangible outcomes.

Thank you for the comment. We added information about cognitive tests in Section 2.

2.1. «The administration of NSC to mice eliminated deviations in solving learning and memorization tasks, such as the Morris water maze and context-dependent novel object recognition [doi: 10.1073/pnas.0901402106].»

2.3. «In an experiment to prevent atrophy of the cholinergic system motor and cognitive functions improved in elderly mice after MSCs transplantation. Conclusions were drawn based on the results of video tracking recording, passage of the water maze and passive avoidance performance [DOI: 10.1002/jnr.23182].»

2.4. «ESC therapy reduced cognitive dysfunction in 5xFad transgenic mice. Spontaneous alteration performance in the Y-maze was used to test the spatial working memory of the mice and contextual fear-conditioning testing was carried out [doi.org/10.1016/j.diff.2009.06.005].»

2.5. «In an experiment with a 5xFad mice line in which five familial Alzheimer's disease transgenes are overexpressed, IPSC injections restored cognitive impairment. The "Y-shaped maze" test was used to test the spatial working memory of mice and contextual fear-conditioning testing was conducted [doi: 10.5966/sctm.2016-0081].»

  1. A section comparing the efficacy, safety profiles, and limitations of NSCs, MSCs, and iPSCs would be beneficial, helping readers grasp the strengths and weaknesses of each approach (Line 25). This comparative analysis would allow readers to see the “bigger picture” of stem cell therapies for AD and make it easier for them to assess which stem cell types might offer the most promise, especially considering the balance of risks and benefits.

Thank you for the comment. A new section 8. Challenges and promises of cell therapy for AD was added, which examined the problems of applying cellular therapy in clinical practice, as well as some clinical trials of certain types of cells in Alzheimer's disease.

  1. Including current clinical applications or trials would ground the review in practical terms (Line 142–143). For example, mentioning specific trials for MSC or iPSC therapies would contextualize the experimental findings, showing readers that these therapies are already under investigation for potential real-world applications. Additionally, discussing the logistical challenges of delivering stem cells to the brain could underscore the need for further innovation in clinical applications.

We discussed some clinical trial in new Section 8. The challenges of delivering stem cells is also discussed in this section as well as other challenges of stem cell therapy in AD.

11. How NSCs migrate post-transplantation can significantly impact their therapeutic efficacy. Discussing whether NSC migration is directed or random (Line 243–245) will help readers understand if transplanted cells localize to AD-affected areas effectively.

Thank you for comment. The information about NSCs migration post-transplantation was added in section 4 «Stem Cell Replacement Therapy».

  1. The manuscript could elaborate on BDNF’s role in AD pathology, especially its ability to improve cognitive function independently of amyloid-beta levels (Line 215–220). A deeper exploration into this amyloid-independent mechanism could present BDNF as a therapeutic target, focusing on cognitive restoration even if amyloid reduction proves difficult.

Thank you for the remark, we included additional information in section 3.

«BDNF and its active form mBDNF play an important role in the development of neurons, their survival and synaptic plasticity by activating the TrkB receptor and triggering various signaling pathways, including MAPK/ERK, PI3K and PLCγ/PKC[doi: 10.1038/nature19766.; doi: 10.1038/mp.2013.134.], which are associated with the activation of the transcription factor CREB[doi: 10.1016/s0896-6273(00)81010-7.]. It, in turn, regulates the activity of genes required for synaptic plasticity. »

  1. The conclusion currently calls for further research but could benefit from specific suggestions, such as refining delivery systems or bioengineering cells for better survival (Line 351–359). Proposing actionable future steps would guide researchers in exploring innovative ways to overcome current limitations in stem cell therapy for AD. Adding targeted recommendations would also position the manuscript as an influential source for shaping future research directions.

We expanded conclusion according to your comment.

«Simplified methods for obtaining safe autologous iPSCs can be a solution to the problems faced by researchers, as a step towards personalized medicine, or rejection of stem cell therapy in favor of paracrine factors produced by these cells. Currently, multiple developments are underway for stem cell therapies to improve targeted delivery or increase their survival rates. However, there are still no clear protocols that would allow ensuring the conduct of therapy with strictly predictable results. »

  1. A flowchart summarizing each stem cell type’s mechanisms (e.g., cell replacement, paracrine effects, exosome action, TNT mitochondrial transfer) would aid comprehension (Line 322). Such visual aids are helpful for complex reviews, allowing readers to quickly understand the scope of therapeutic actions associated with each cell type. Including a summary chart or figure would also enhance the review’s appeal as a comprehensive reference, especially for researchers or students visualizing mechanisms at a glance.

Thank you for the remark. The manuscript was supplemented with a diagram that condenses the information from the review on the primary mechanisms of action of different types of stem cells in Alzheimer's treatment.

Reviewer 3 Report

Comments and Suggestions for Authors

In the article entitled “Proposed Mechanisms of Cell Therapy for Alzheimer's Disease” the authors give an extensive review of the various types of stem cells, including embryonic stem cells and induced pluripotent stem cells as a potential treatment for Alzheimer's disease through neuronal regeneration and cognitive improvement.

It would be important to address the following topics.

1.     This is a very interesting review and it would be appropriate to go into more depth on the mechanisms of action of stem cells.

2. It is also important to address how stem cells can reduce neuroinflammation.

3. It would be important to further describe how stem cells could reduce Tau hyperphosphorylation and beta-amyloid aggregation.

4.     The proposed mechanism would be with FDA-approved drug interaction.

5.     What advantages would stem cell treatment have over the use of Lecanemab which is the antibody recently employed in the treatment of the disease.

6.     Also mention the possible cost-benefit of the treatment.

7.     Side effects of the treatment that have been observed in animals.

8.     Ethical aspects of the treatment.

Author Response

Dear Reviewer,

We would like to express our sincere gratitude for reviewing our work. Your feedback and constructive comments are highly appreciated.

  1. This is a very interesting review and it would be appropriate to go into more depth on the mechanisms of action of stem cells.

Thank you for your comment. We added information about the mechanisms of each type of stem cells in Alzheimer's disease in section 2. Sections 4, 5, 6 and 7 were supplemented with the results of recent studies, and the discussion of the mechanisms of action was expanded. We also added a diagram that condenses the information from the review.

  1. It is also important to address how stem cells can reduce neuroinflammation.

We thank you for your fair comment. In chapter 1, we added information about the various ways in which stem cells influence neuroinflammation. Changes in the production of specific pro-inflammatory mediators after treatment with stem cells were also specified.

  1. It would be important to further describe how stem cells could reduce Tau hyperphosphorylation and beta-amyloid aggregation.

Thank you for the remark. We supplemented the manuscript with the disscution about Tau hyperphosphorylaton and beta-amyloid aggragation in Sections 5 and 6.

Section 5. «Studies also have shown that an increase in BDNF levels in Alzheimer's disease leads to a reduction in the production of toxic Aβ [https://doi.org/10.1111/jnc.14034] through alpha-secretase processing of APP, as well as inhibition of tau phosphorylation [https://doi.org/10.1016/j.neuropharm.2021.108737].»

Section 6. «Various studies have shown that MSC exosomes enhance angiogenesis, promote functional recovery and neurovascular remodeling [10.1186/s13287-020-01834-0; https://doi.org/10.1038/jcbfm.2013.152], while NSC exosomes often inhibit neuroinflammation and microglia activation [DOI: 10.1186/s13195-021-00791-x; DOI: 10.1111/jnc.15001]. Gao et al.'s latest research has demonstrated that NSC-EV have a better effect in cognitive tests (water maze test, fear conditioning test), show a more pronounced inhibitory effect on Aβ1-42 accumulation and Tau phosphorylation, and have better therapeutic effects on restoring dendrite length and density of dendritic spines compared to MSC-EV in the mouse AD model [s41392-023-01436-1].»

  1. The proposed mechanism would be with FDA-approved drug interaction.

We supplemented Section 8 with «Today the FDA has fast track designation status to the Lomecel-B drug for the treatment of mild Alzheimer's disease, developed by Longeveron. The drug, consisting of mesenchymal stem cells obtained from the bone marrow of healthy donors, has shown positive results in studies, improving cognitive function, and slowing the loss of brain volume in patients. Amyloid-related imaging abnormalities (ARIA) of brain were not detected in patients as a result of taking the drug. We believe that the cell therapy will soon be approved by the FDA.»

  1. What advantages would stem cell treatment have over the use of Lecanemab which is the antibody recently employed in the treatment of the disease.

We added information in Section 8. «Ð’ased on data from the IIa phase of clinical trials of Lomecel-B, no abnormalities of brain imaging on MRI associated with amyloid deposition (ARIA) were detected in patients as a result of taking the drug in comparison with Lecanemab.  Lecanemab may cause ARIA, that most often manifests itself as temporary swelling of the brain, which usually passes over time and may be accompanied by small hemorrhages in the brain or on its surface, although some people may experience symptoms such as headache, confusion, dizziness, vision changes, nausea and seizures.»

  1. Also mention the possible cost-benefit of the treatment.

We added information in Section 8«Cultivation of MSCs and their expansion in the laboratory has been a routine task that does not require expensive reagents and a lot of effort in comparison with antibodies, which require high reagent costs to be obtained.»

  1. Side effects of the treatment that have been observed in animals.

Section 2, titled «Stem Cells in AD Therapy», now includes new data on the disadvantages of using each type of cells for therapy. A new section 8 was added, which is called «Challenges and promises of cell therapy for AD». It explores the issues associated with implementing cellular therapy in clinical practice. The section also covers some clinical trials of particular types of cells used in treating Alzheimer's disease.

2.1. «One of the main disadvantages of the NSC therapy method is poor viability. In many cases, losses can be as much as 90%. Therefore, NSC replacement requires additional conditions to maintain stem cell survival [doi: 10.1021/bc400005m, doi: 10.1016/j.expneurol.2013.04.001].»

2.2. «Despite the fact that embryonic GPC look promising as potential therapeutic vectors, they have limitations in their initial number and ability to reproduce. Because of this, it is required to periodically receive new GPCs from donor tissues [DOI: 10.1126/science.1218071].»

2.3. «At the same time, MSC therapy is not flawless. Studies have noted a low rate of neuronal differentiation, and moreover, it depends on the microenvironment of the host brain. The differentiation pathway may differ by the microenvironment of one organism to another, which plays an important role in determining the fate of transplanted MSC [doi.org/10.1046/j.1440-1789.2003.00496.x].»

2.4. «Despite the obvious advantages of pluripotency, it is becoming one of the main problems of using ESC. In many studies, researchers have noted the uncontrolled differentiation of this type of stem cells and the formation of teratomas [doi: 10.1038/nbt726 , doi: 10.1016/S0002-9440(10)62488-1].»

2.5. «It is worth noting that iPSC requires increased attention, especially at the reprogramming stage due to the possible pathological phenotype [doi: 10.1186/s12864-015-1262-5]. iPSCs also tend to differentiate according to the type of donor cells and may inherit epigenetic features from the original ones [https://doi.org/10.1007/978-4-431-55966-5_1].»

  1. Ethical aspects of the treatment.

We supplemented Section 8 with «Cell therapy raises ethical issues such as the source of stem cells and the donor's and patient's sufficient awareness. Human ESCs are obtained from a 5–7-day-old blastocyst. In many countries, research on human embryos is limited. Due to the risk of creating a business for production and the phenomenon of women's exploitation, it is impossible to create the possibility of free paid donation. Nevertheless, donor cells donated on a voluntary basis are currently the most acceptable way to obtain ESCs. It follows that in order to use both "unnecessary to parents" blastocysts and to receive donor ones, it is necessary to create a strict international regulation according to which ESCs will be recorded and all accompanying documentation. It is necessary to determine what amount of information should be provided to a patient undergoing stem cell treatment. You should not mislead seriously ill patients and offer them unproven methods of therapy.»

Round 2

Reviewer 1 Report

Comments and Suggestions for Authors

The manuscript was appropriately revised and can be accepted. Please change the Conclusion part to section 9.

Reviewer 2 Report

Comments and Suggestions for Authors

The authors succeeded in responding to all my previous comments and improved their manuscript to the acceptance level.

Reviewer 3 Report

Comments and Suggestions for Authors

Thanks to the authors for their replies. The manuscript improved considerably with the suggested changes.